# Effects of Carbohydrate Restriction on Body Weight and Glycemic Control in Individuals with Type 2 Diabetes: A Randomized Controlled Trial of Efficacy in Real-Life Settings

**DOI:** 10.3390/nu14245244

**Published:** 2022-12-09

**Authors:** Philip Weber, Mads N. Thomsen, Mads Juul Skytte, Amirsalar Samkani, Martin Hansen Carl, Arne Astrup, Jan Frystyk, Jens J. Holst, Bolette Hartmann, Sten Madsbad, Faidon Magkos, Thure Krarup, Steen B. Haugaard

**Affiliations:** 1Department of Endocrinology, Copenhagen University Hospital Bispebjerg, 2400 Copenhagen, Denmark; 2Department of Forensic Medicine, University of Copenhagen, 2100 Copenhagen, Denmark; 3Department of Obesity and Nutritional Sciences, Novo Nordisk Foundation, 2900 Hellerup, Denmark; 4Department of Clinical Medicine, Aarhus University, 8200 Aarhus, Denmark; 5Department of Endocrinology, Odense University Hospital, 5000 Odense, Denmark; 6NNF Centre for Basic Metabolic Research, University of Copenhagen, 2200 Copenhagen, Denmark; 7Department of Biomedical Sciences, University of Copenhagen, 2200 Copenhagen, Denmark; 8Department of Endocrinology, Copenhagen University Hospital Amager Hvidovre, 2650 Hvidovre, Denmark; 9Department of Nutrition, Exercise and Sports, University of Copenhagen, 2200 Copenhagen, Denmark; 10Institute of Clinical Medicine, University of Copenhagen, 2200 Copenhagen, Denmark

**Keywords:** macronutrients, type 2 diabetes, carbohydrate restriction, glucose metabolism, lipid metabolism

## Abstract

A fully provided, hypocaloric, carbohydrate-reduced high-protein (CRHP) diet compared to a hypocaloric conventional diabetes (CD) diet for 6 weeks improved glycemic control to a greater extent in face of an intended 6% weight loss in individuals with type 2 diabetes mellitus (T2DM). The present 24-week extension of that study reports on the efficacy of CRHP and CD diets in a real-life setting. Sixty-five individuals with T2DM who completed the initial 6-week fully provided diet period (% energy from carbohydrate, protein, and fat was 30/30/40 in CRHP, and 50/17/33 in CD) continued a free-living, dietician guided 24-week period of which 59 individuals completed. The CRHP compared to CD group reported a 4% lower carbohydrate intake and had higher urea excretion by 22% (both *p* ≤ 0.05) at week 30, suggesting less difference in carbohydrate and protein intake between groups during the 24-week extension compared to week 6. The loss of body weight during the initial 6 weeks was maintained in both groups during the 24-week extension (−5.5 ± 4.5 and −4.6 ± 4.8 kg) as well as HbA_1c_ (−8.4 ± 6.2 and −8.4 ± 6.9 mmol/mol) with no significant differences between groups. The additional benefits on glucoregulation harnessed by carbohydrate restriction under full diet provision for 6 weeks combined with titrated weight loss could not be maintained in a real-life setting of self-prepared diet aiming on similar diets for 6 months.

## 1. Introduction

Weight loss improves glycemic control and cardiovascular risk factors in patients with type 2 diabetes mellitus (T2DM) [1,2]. However, what is less clear is to what extent, the macronutrient composition of the diet influences glycemic control and the risk of diabetic complications independent of its possible effects on body weight [3]. Dietary intake of carbohydrate is decisive for postprandial glucose excursions, worsening glucoregulation and increases glycated hemoglobin (HbA_1c_) [4,5]. However, studies investigating low-carbohydrate diets in the management of T2DM are heterogeneous in design, e.g., patient characteristics, type of dietary intervention, extent of carbohydrate restriction, and duration, which may explain the inconsistent results [6,7]. Compliance and adherence to dietary protocol are difficult to validate and depends on the type of dietary intervention and its duration, possibly blurring the potential long-term effect of carbohydrate restriction on glycemic control [6].

This raises the question of feasibility of carbohydrate restriction as a therapeutic regimen in the treatment of T2DM. We have previously reported a significant reduction in HbA_1c_ after 6 weeks on a fully provided, weight-maintaining carbohydrate-reduced high-protein (CRHP) diet compared to an isocaloric conventional diabetes (CD) diet in patients with T2DM [8]. We further observed that the beneficial metabolic effects of the CRHP diet were maintained, or further improved, during the subsequent 24 weeks, when patients transitioned to a free-living setting and had to select and prepare the CRHP diet by themselves, with frequent dietitian guidance (every other week) [9,10]. More recently, we demonstrated that the same CRHP diet extended the beneficial metabolic effects of weight loss when applied during calorie restriction in a fully controlled setting for 6 weeks [11]. In this follow-up study, we report on the results of the subsequent 24 weeks of free-living diet intervention to investigate if the additive metabolic effect of diet-induced weight loss and carbohydrate reduction could be maintained when patients self-prepared diets with the aid of dietitian guidance.

## 2. Materials and Methods

### 2.1. Study Design

The initial 6-week fully provided diet period has been described earlier [11]. In this paper we report on the subsequent 24-week self-selected diet phase, conducted as an open-label, parallel trial with two study arms. Participants were randomized in a 1:1 ratio in random block sizes by a third-party using R (extension package ‘blockrand’, Version 3.6.0; R, Boston, MA, USA) to the CRHP or the CD diet. Upon completion of the initial 6-week period, during which diets were fully provided free-of-charge, participants transitioned to a 24-week free-living period during which they had to purchase the food items and prepare the diets themselves.

The original study population consisted of men and women (>18 years of age) with overweight or obesity (body mass index > 25 kg/m^2^), T2DM as defined by current American Diabetes Association guidelines, and HbA_1c_ between 48 mmol/mol (6.5%) and 97 mmol/mol (11%) [12]. Participants were excluded if suffering from critical illness including earlier or ongoing severe cardiac or gastrointestinal illness, chronic kidney disease stage 4 or 5 (estimated glomerular filtration rate [eGFR] <30 mL/min/1.73 m^2^) or cancer. Treatment with sulfonylureas, sodium glucose cotransporter 2 (SGLT2) inhibitors, systemic corticosteroids, or injectable glucose-lowering medications such as insulin and glucagon-like peptide 1 (GLP-1) analogues excluded subjects from participation. During the follow-up period, all medications (and changes in them) prescribed by the patients’ general practitioners were allowed. Participants were recruited from the greater Copenhagen region. All assessments were conducted at the research unit of the Department of Endocrinology at University Hospital at Bispebjerg and Frederiksberg in Copenhagen. All study activities took place from January 2019 to December 2020. Written and oral consent was obtained from all participants prior to the study. The study is registered with www.clinicaltrials.gov (NCT03814694).

### 2.2. Intervention

During the 24-week follow-up, study participants were encouraged to continue their allocated diets and maintain the weight loss achieved during the initial 6-week food provision period. The CRHP diet provided 30 percent of energy (E%) from carbohydrates, 30 E% from protein, and 40 E% from fat, whereas the CD diet provided 50 E% from carbohydrates, 17 E% from protein, and 33 E% from fat. Participants were provided with recipes and dietitian guidance to help them adhere to the allocated diets. Three consulting sessions lasting 90, 60 and 60 min at weeks 7, 14 and 22, respectively, were conducted by a clinical dietitian in group-based sessions of 4–7 participants; advice focused on dietary adherence and weight loss maintenance. Reducing dietary saturated fatty acid intake at <10 E% and prioritizing carbohydrates low in glycemic index and rich in fiber were recommended to both groups, in accordance with current dietary guidelines of the Diabetes and Nutrition Study Group of the EASD [13]. The same dietitian was responsible for both study arms to ensure equivalency of the dietetic advice. Participants registered their diet for 3 consecutive days, including 2 working days and 1 weekend day, using an online diet registry tool (www.madlog.dk, MADLOG ApS, Kolding, Denmark) before intervention (week 0) and at the end of the trial (week 30).

### 2.3. Blood Samples

Blood samples were collected at baseline (week 0), at the end of the fully provided diet period (week 6) and at study termination (week 30). Blood was drawn from an antecubital vein and distributed to tubes coated with EDTA for plasma and silicon dioxide for serum preparation. Plasma tubes (immediately after collection) and serum tubes (30 min after collection) were centrifuged at 2000 g for 10 min (Centrifuge 5702 R, Eppendorf Nordic A/S, Hørsholm, Denmark). To assess glycemic control, HbA_1c_ was measured in whole blood on a Tosoh Automated Glycohemoglobin Analyzer HLC-723G8 (Tosoh Corp., Tokyo, Japan) and fasting plasma glucose was measured on a YSI 2300 StatPlus Glucose Analyzer (YSI Inc., Yellow Springs, OH, USA). To evaluate lipid profile, serum was analyzed for total cholesterol, high-density lipoprotein (HDL) cholesterol, low-density lipoprotein (LDL) cholesterol and triglycerides using enzymatic colorimetric methods, and non-esterified fatty acids (NEFA) were measured using spectrophotometry (CHOL2, HDLC4, LDLC3, TRIGL, and NEFA-HR (2) kits on a COBAS c6000, Roche Diagnostics, Basel, Switzerland). Insulin and C-peptide were measured by using immunoassays (Elecsys Insulin and Elecsys C-peptide kits; COBAS c6000). Apolipoprotein A-I (ApoA-I) and apolipoprotein B (ApoB) were measured using immunoturbidimetric analyses (APOBT and APOAT kits; COBAS c6000). Non-HDL cholesterol was calculated by subtracting HDL cholesterol values from total cholesterol values. Kidney function was evaluated based on eGFR calculated from serum cystatin C following the equation suggested by Grubb et al. [14]. Cystatin C was measured using particle-enhanced immunoturbidimetric analysis (CYSC2 kit; COBAS c6000).

### 2.4. Urine Samples

Twenty-four-hour urine samples were collected at baseline (week 0), during the last week (week 5) of the initial 6-week phase, and twice during follow-up (weeks 22 and 30). Participants were supplied with plastic containers and instructed to collect urine for 24 h and to keep it refrigerated at 4° Celsius in thermal boxes. Urine samples were delivered to the lab within 24 h of completed urine collection, where total weight was recorded and density was measured by using a refractometer (PAL-10S, ATAGO^®^, Tokyo, Japan). Urine urea nitrogen (UUN) was analyzed using a kinetic test with urease and glutamate (UREAL kit; COBAS c6000) and albumin was measured by using immunoturbidimetry (ALBT2 kit; COBAS c6000). All biological samples were stored at −80° Celsius until analysis.

### 2.5. Statistical Analysis and Calculations

Changes in endpoints were determined both relative to baseline (combined 30-week intervention, p1) and relative to the end of the initial 6-week phase (isolated 24-week follow-up, p2). To assess p1 we used a baseline-adjusted constrained linear mixed effect model assuming unstructured covariance pattern to account for repeated measures. The same model was used to investigate differences over time within each diet group, for exploratory purposes and to evaluate differences between groups in macronutrient intake and urea excretion. To assess p2 we used a similar linear mixed effect model which, however, did not adjust for any differences at week 6. In both types of analyses, residuals were evaluated for normal distribution, and log transformation was applied when necessary. All statistical analyses were made in R version 3.5.2 (R foundation for statistical computing, Boston, MA, USA) and a level of statistical significance of 5% was used to reject the null hypothesis. A sample size of 80 participants was estimated based on the primary and secondary outcomes (HbA_1c_ and hepatic fat fraction) in the initial 6-week study and a maximum drop-out rate of 20% [11]. Insulin resistance (HOMA2-IR) and beta cell function (HOMA2-B%) were calculated with the updated HOMA2 calculator (version 2.2.3; www.dtu.ox.ac.uk/homacalculator/) based on fasting plasma glucose, C-peptide and insulin [15]. Fisher’s exact test was used to test for significance of changes in anti-hyperglycemic, lipid-lowering and anti-hypertensive medications between diet groups.

## 3. Results

### 3.1. Participants

Seventy-two participants commenced the initial 6-week fully provided diet period; 2 withdrew before intervention started, 3 discontinued during the initial phase, and 2 more declined continuation in the follow-up. Among the 65 participants who continued to the 24-week follow-up, 59 completed the entire study (30 weeks). By an intention-to-treat principle, all participants were encouraged to attend every assessment regardless of diet adherence and previous number of visits. Descriptive values presented include only participants who completed the entire study and provided all blood samples, while all available data were included in the linear mixed models to facilitate intention-to-treat analysis. Baseline characteristics are listed in Table 1. Participation flow diagram is shown in Figure 1. Adverse events were registered during the 6-week food provision period only. Thirteen participants (5 in the CD group and 7 in CRHP group) experienced mild constipation symptoms, while one participant in the CRHP group experienced moderate constipation. Two participants in each group experienced diarrhea. Three participants (1 in the CD group and 2 in the CRHP group) experienced dizziness. One participant experienced a transient episode of sweating and increased levels of plasma creatinine; no underlying medical cause was identified [11].

### 3.2. Self-Prepared Diet

According to the food records collected at study termination (Table 2), total daily energy intake was increased but non-significantly in the CRHP group compared to the CD group, with an estimated difference of 371 (95% CI: −606; 1347, *p* = 0.5) kJ/day. The daily intake of carbohydrate was significantly reduced in the CRHP group compared with the CD group by −4.4 (95% CI: −7.1; −1.7, *p* = 0.002) E%, but this should be evaluated in the contexts that the difference in carbohydrate supplied in the full food provision period was 20 E%. The intake of monounsaturated fatty acids increased by 1.4 (95% CI: 1.1; 1.8, *p* = 0.006) g/day in the CRHP group, whereas no other significant differences were measured by the food records.

At baseline, UUN excretion was similar in the CRHP group relative to the CD group (6 [95% CI: −18; 49, *p* = 0.4]%). At week 5 (last week of initial phase), it was 66 (95% CI: 42; 95, *p* < 0.001)% higher in the CRHP group compared to the CD group. At weeks 22 and 30 (during the follow-up), UUN excretion in the CRHP group declined, compared to the food provision period, but remained greater than in the CD group by 27 (95% CI: 4; 56; *p* = 0.02)% and 22 (95% CI: 0; 50; *p* = 0.05)%, respectively (Figure 2) suggesting a persisting larger protein intake in the CRHP group during follow-up.

The intake of protein (in E%) provided by the food records correlated significantly with total urea excretion (pearson’s *r* = 0.48, *p* < 0.001) in a combined analysis of all measurements before trial initiation, at week 5 during the food provision period and at week 30 after follow-up, validating the self-reported food records.

### 3.3. Anthropometrics

Body weight decreased equally in both groups during the initial 6-week period (CRHP: −5.8 [SD: ±1.9] kg and CD: −5.8 [SD: ±2.3] kg). By the end of follow-up (week 30), body weight remained significantly lower compared with baseline by −5.5 (SD: ±4.5) kg in the CRHP group and by −4.6 (SD: ±4.8) kg in the CD group with no difference between groups (Table 3).

### 3.4. Glucose Metabolism

All markers of glucose metabolism improved at the end of the study (week 30) compared with baseline, however, the improvements were independent of diet allocation (Table 3). The initial 6 weeks of full diet provision led to a significantly greater reduction in HbA_1c_ by 1.9 mmol/mol in the CRHP than the CD diet group [11]. After the entire 30-week intervention, HbA_1c_ remained significantly lower compared with baseline, but the extent of reduction was not different between groups (CRHP: −8.4 [SD: ±6.2] and CD: −8.4 [SD: ±6.9] mmol/mol, *p* = 0.7). Fasting glucose, insulin, C-peptide, HOMA-IR and HOMA-B% were not different between diet groups.

### 3.5. Lipid Profile

Total cholesterol, non-HDL cholesterol, LDL cholesterol and triglycerides decreased in both groups during the initial 6-week period, and the beneficial effect persisted for triglycerides at week 30 (Table 3). HDL cholesterol increased significantly in both groups at the end of follow-up with no difference between diets (CRHP: 0.2 [SD: ±0.2] mmol/l and CD: 0.2 [SD: ±0.1] mmol/l, *p* = 0.2). NEFA decreased significantly and to the same extent in both groups at the end of the study compared with baseline (CRHP: −0.2 [IQR: −0.3; −0.1] mmol/L and CD: −0.3 [IQR: −0.4; −0.1] mmol/L, *p* = 0.8); this change was not apparent during the initial 6-week phase. ApoA-I decreased in both groups during the initial 6-week period but increased significantly in both groups during the subsequent 24-week follow-up. ApoB decreased significantly in both groups at week 6 but this was no longer apparent at week 30.

### 3.6. Renal Function

No difference in eGFR was observed between diets after the initial 6-week phase, but eGFR was slightly reduced in both groups at study conclusion (CRHP: −5.1 [SD: ±4.6] and CD: −5.2 [SD: ±7.0] mL/min/1.73 m^2^; *p* = 0.9). Urine albumin concentration was unchanged throughout the entire study (Table 3).

### 3.7. Medication Changes

A few participants had their anti-hyperglycemic, lipid-lowering or anti-hypertensive medication increased or decreased by their general practitioners during the free-living follow-up (Table 4). Five participants in the CD diet group and 4 in the CRHP diet group had their anti-diabetic medication increased during follow-up, while 5 (CD) and 7 (CRHP) participants had their medication decreased. Three participants in the CD group and none in the CRHP group had their lipid-lowering medication increased, and no participant had their medication decreased. Anti-hypertensive medication was increased in 3 participants in both diet groups, while 1 and 2 participants in the CD and CRHP diet groups, respectively, had their medication decreased.

## 4. Discussion

In this study, we observed that adherence to a CRHP diet was reduced when individuals with T2DM transitioned from a highly controlled period of full diet provision to a free-living period of self-selected and self-prepared diets, thereby likely attenuating the beneficial metabolic effects initially harnessed from carbohydrate restriction. Thus, the improvements in glucose metabolism and circulating triglycerides that occurred with a CRHP feeding period during the initial 6 weeks of the study [11] were no longer apparent after the subsequent 24 weeks of follow-up. After participants transitioned to free-living conditions with group dietitian guidance every other month, weight loss and metabolic improvements including those in glycemic control were maintained to the same extent in both groups. This finding is in accordance with long-term studies on self-prepared diets with intended difference in macronutrient compositions [16,17,18] and reinforce the primary importance of weight loss rather than dietary macronutrient composition in the long-term management of T2DM.

In a previous study [8] with a crossover design, improvements in glycemic control were substantial on a weight-maintaining CRHP diet relative to an isocaloric CD diet. In that study, the difference between diets of reduction in HbA_1c_ was 5.4 mmol/mol after 6 weeks of full diet provision [8], and during an open-label 6-month extension when all patients were free-living and self-preparing the CRHP diet, a reduction of 7 mmol/mol was found relative to baseline in the absence of major weight loss [9]. In the first 6 weeks of the present study, when hypocaloric diets led to the same modest (~6%) weight loss in each diet group, the difference in HbA_1c_ between diets was small but significant, at 1.9 mmol/mol [11]. This modest difference in HbA_1c_ between diets supports the notion that greater carbohydrate restrictions convert to larger improvements in HbA_1c_ [19,20]. However, during the present extension study this initial difference in HbA_1c_ between diets was lost during follow-up. We speculate that this is caused by a declining adherence to the CRHP diet, as reflected by the food records and the UUN excretion data. Collectively, results from these studies emphasizes that weight loss is a very strong modifier of HbA_1c_ [2] and likely masks additional effects of dietary macronutrient shifts if the change is only modest. This implies that high diet adherence may be critical to achieve the additive beneficial effects of carbohydrate restriction on glycemic control, when applied in concert with weight reduction.

Individuals with obesity and T2DM are predisposed to non-alcoholic fatty liver disease (NAFLD) [21], known as a driver of hepatic insulin resistance and excessive hepatic secretion of triglyceride [22,23]. Individuals with obesity and NAFLD exhibit lower inhibition of lipolysis in peripheral adipose tissue leading to increased NEFA release and circulating NEFA concentrations [24,25]. We recently demonstrated that liver fat can be mobilized through carbohydrate restriction, beyond the effects of weight loss [8,11]. In the present study, we observed a significant decrease in fasting NEFA in both diet groups at the end of follow-up, possibly reflecting improved adipose tissue insulin sensitivity and enhanced suppression of lipolysis, which was not apparent at the end of the initial 6-week period. We made a similar observation previously, during 6 months on a self-selected weight-maintaining CRHP diet, when fasting and postprandial NEFA concentrations were significantly lower at the end of the follow-up study than at the end of the initial 6-week fully provided diet phase [10]. Thus, the duration of carbohydrate restriction may be important to facilitate the lowering of NEFA and, consequently, reduction in lipotoxicity.

HDL particles facilitate the reverse transport of excess cholesterol from peripheral tissues to the liver [26] and possess multiple atheroprotective functions [27]. The effect of carbohydrate restriction on plasma levels of HDL cholesterol is not clear because of conflicting results [28,29]. In the initial 6 weeks of this trial, we observed a decrease in HDL cholesterol in both groups—likely attributed to weight loss—followed by a rebound, which led to an overall increase in HDL cholesterol for the entire 30-week study. Similar results have been reported for ApoA-I, which is an independent cardiovascular risk predictor [30] that reflects the number of HDL particles [31]. The combined increase in serum HDL cholesterol and ApoA-I during the 6-month follow-up in both groups in the present study is in agreement with our previous observations during a weight-maintaining CRHP diet [10] and indicates that longer periods of time may be required for this cardiovascular benefit to manifest in response to carbohydrate restriction.

Most meta-analyses find beneficial metabolic effects of low-carbohydrate diets in the short-term, but have failed to demonstrate benefits in the long-term, which partly may be explained by the lack of long-term studies and diminishing adherence to carbohydrate restriction set out by the studies [6,19,32]. Adherence to the prescribed diet is likely a major factor in long-term dietary intervention studies [33]. The methods used to facilitate dietary adherence are heterogeneous between studies, making the ideal approach difficult to identify [6]. In this study, we hypothesized that participants who were provided all meals during the first 6 weeks would be able to translate this experience into a real-life setting and adhere to the prescribed diets under free-living conditions for 24 additional weeks. However, as evidenced by the food records and the UUN excretion data, adherence to lower carbohydrate and greater protein intake was not successful and declined over time. The low frequency of meetings with the research dietitian (once every other month) may also have resulted in macronutrient targets not being met [19,32] as dietitian counselling once weekly or every other week is suggested as an effective behavioral approach to achieve satisfactory adherence to a diet plan [34,35]. In the 6-month follow-up of our previous study, patients met with the research dietitian every other week and their reported macronutrient intake was very close to the targeted one, which was likely instrumental in the maintenance of CRHP-induced metabolic improvements [9,10]. The findings from these studies, together with the present results, may imply that a tightly controlled diet induction period does not translate into better long-term dietary adherence in a real-world setting in the absence of frequent dietitian counselling.

This follow-up study has several limitations; open-label design which is a natural consequence of dietary studies. The short study duration, group-based rather than individual dietetic advice, and the relatively low frequency of dietitian guidance were chosen to comply with the limited study resources. The lack of accurate methods to account for changes in medication during follow-up may have clouded differences in the metabolic effect between dietary groups. The strengths of this study include a relatively large study population with high completion rate, the standardized dietitian guidance between groups, and the objective evaluation of dietary adherence by multiple UUN measurements. Additionally, the two-phase design of the study, consisting of an initial intensive 6-week period with full diet provision, with matched weight loss (6%), followed by a free-living 24-week period of weight loss maintenance, during which diets were self-prepared, has only been sparingly investigated and could be considered as an implementation strategy.

## 5. Conclusions

The additive metabolic effects of carbohydrate reduction from weight loss were lost in the present extension study, when participants transitioned from full food provision to a real-life setting. This contrasted our previous findings from carbohydrate restriction without accompanying weight loss, maybe because participants were unable to simultaneously adhere to the allocated diet pattern and keep their weight reduction. Maintenance of weight losses was shown to be instrumental in improving glycemic control in T2DM. Future studies must address how to improve long-term adherence to carbohydrate-restricted dietary regimens to determine the possible positive effect of such diets in improving several pathophysiological traits of T2DM.

## Figures and Tables

**Figure 1 nutrients-14-05244-f001:**
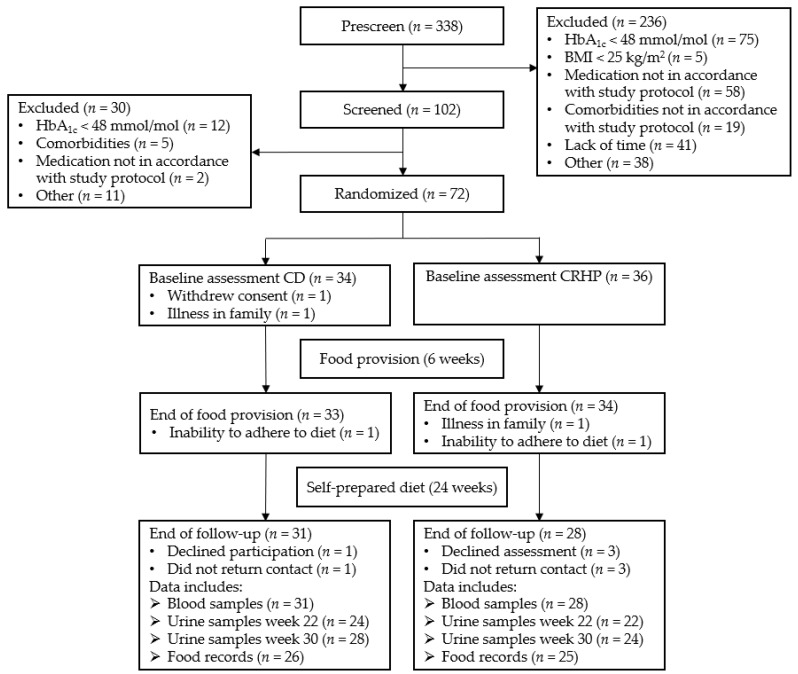
Participant flow diagram.

**Figure 2 nutrients-14-05244-f002:**
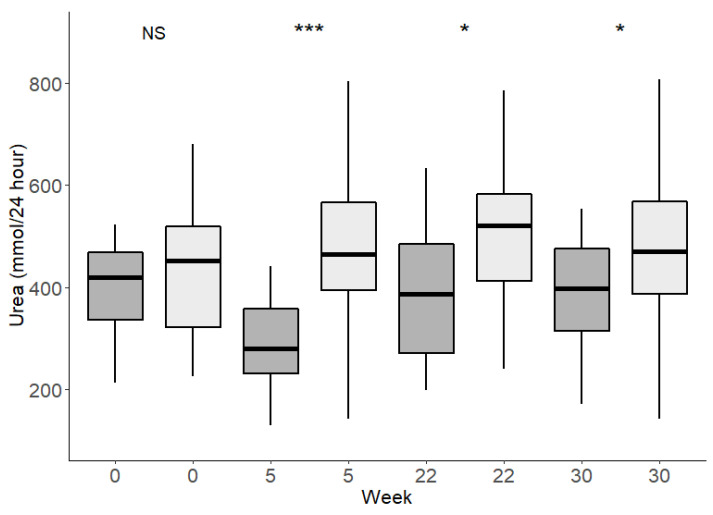
Twenty-four-hour UUN excretion. Dark boxes represent CD group and light boxes represent CRHP group. Before intervention start (week 0), during food provision (week 5), during and at end of follow-up (week 22 and 30). Between diet groups significance level; * *p* < 0.05, *** *p* < 0.001, NS = not significant.

**Table 1 nutrients-14-05244-t001:** Baseline Characteristics.

Characteristics	CD	CRHP
Participants (male)	31 (15)	28 (17)
Age (years)	67.0 (±8.4)	66.9 (±6.9)
Duration of T2DM (years)	7.7 (3.2; 10.0)	8.2 (3.9; 12.7)
Height (cm)	170.7 (±11.8)	170.6 (±8.3)
Body weight (kg)	97.4 (±25.9)	98.8 (±13.4)
Body mass index (kg/m^2^)	33.0 (±5.2)	34.0 (±4.7)
HbA_1c_ (mmol/mol)	57.7 (±7.8)	57.0 (±6.7)
Medication		
No hypoglycemic agents	11	7
1 hypoglycemic agent	17	13
2 hypoglycemic agents	3	8
Lipid lowering medication	24	24
Anti-hypertensive medication	21	22

Data are shown as mean (±SD) or median (quartile 1; quartile 3). Medication use and sex are indicated as frequency counts.

**Table 2 nutrients-14-05244-t002:** Daily dietary intake of energy and macronutrients assessed by 3-day food records.

	Baseline				Week 30					
	CRHP	*n*	CD	*n*	CRHP	*n*	CD	*n*	Between-Diet Difference (95% CI)	*p*-Value
Total energy (kJ)	8604 (±2789)	35	8488 (±2488)	35	8296 (±2904)	25	7273 (±2067)	26	371 (−606; 1347)	0.5
Total energy (kcal)	2065 (±669)	35	2037 (±597)	35	1991 (±697)	25	1746 (±496)	25	88.5 (−145; 322)	0.5
Protein (E%)	18.6 (±4.3)	35	18.6 (±3.4)	35	21.0 (±5.4)	25	19.8 (±4.7)	26	2.2 (−0.3; 4.6)	0.09
Carbohydrate (E%)	43.4 (±7.1)	35	41.4 (±6.9)	35	38.8 (±5.9)	25	42.3 (±6.1)	26	−4.4 (−7.1; −1.7)	0.002
Fibre (g/day)	25.1 (±8.8)	35	23.1 (±9.0)	35	23.4 (±8.3)	25	26.1 (±13.6)	26	−3.3 (−9.2; 2.6)	0.3
Fat (E%)	34.9 (±6.2)	35	37.2 (±6.4)	35	36.2 (±7.3)	25	35.4 (±4.3)	26	1.4 (−1.5; 4.4)	0.3
Saturated fatty acids (g/day)	22.3 (17.2; 33.8)	35	25.8 (17.2; 33.8)	34 ^A^	22.3 (16.3; 28.3)	25	22 (16.8; 25.6)	26	2.7 (−1.4; 6.8)	0.2
Monounsaturated fatty acids (g/day)	18.7 (12.8; 23.5)	35	21.8 (13.9; 27.0)	34 ^A^	19.0 (13.7; 22.7)	25	13.8 (10.1; 18.7)	26	1.4 (1.1; 1.8)	0.006
Polyunsaturated fatty acids (g/day)	8.5 (5.7; 10.0)	35	8.6 (6.9; 11.1)	34 ^A^	8.2 (5.5; 11.2)	25	7.1 (5.4; 9.4)	26	1.3 (−1.0; 1.7)	0.1
Alcohol (E%)	0.0 (0.0; 3.2)	35	0.0 (0.0; 2.9)	35	0.23 (0.0; 4.2)	25	0.0 (0.0; 4.2)	26	1.0 (−0.5; 2.5)	0.2

Descriptive statistics are expressed as mean (±SD) or median (quartile 1; quartile 3). A linear mixed model was used to estimate the marginal mean difference (95% confidence interval) in dietary intake (CRHP vs. CD) after adjusting for baseline. Residuals were tested for normal distribution and log transformation was done when deemed appropriate. ^A^ = one food record had missing values from fatty acid distribution due to a technical issue. All available data are included in both descriptive and inferential statistics.

**Table 3 nutrients-14-05244-t003:** Anthropometric and metabolic markers.

	CD			CRHP			p1	p2
	Baseline (*n* = 31)	Week 6 (*n* = 31)	Week 30 (*n* = 31)	Baseline (*n* = 28)	Week 6 (*n* = 28)	Week 30 (*n* = 28)		
Anthropometrics								
Body weight (kg)	97.4 (±25.9)	−5.8 (±2.3) ^†††^	−4.6 (±4.8) ^†††^	98.8 (±13.4)	−5.9 (±1.9) ^†††^	−5.5 (±4.5) ^†††^	0.5	0.5
Body mass index (kg/m^2^)	33.0 (±5.2)	−2.0 (±0.6) ^†††^	−1.6 (±1.7) ^†††^	34.0 (±4.7)	−2.0 (±0.6) ^†††^	−1.9 (±1.6) ^†††^	0.6	0.6
Glucose metabolism								
HbA_1c_ (mmol/mol)	57.7 (±7.8)	−7.2 (±4.1) ^†††^	−8.4 (±6.9) ^†††^	57.0 (±6.7)	−9.1 (±3.6) ^†††^	−8.4 (±6.2) ^†††^	0.7	0.4
Fasting glucose (mmol/L)	8.5 (7.9; 10.0)	−1.5 (−2.3; −1.1) ^†††^	−1.1 (−1.7; −0.5) ^†††^	8.3 (7.5; 9.8)	−1.7 (−2.7; −1.0) ^†††^	−1.0 (−1.7; −0.3) ^†††^	0.8	0.7
Insulin (pmol/L)	124.7 (83.2; 183.0)	−27.7 (−54.2; −9.0) ^†††^	−33.3 (−57.6; −2.8)^††^	120.3 (96.1; 151.5)	−24.0 (−50.9; −9.5) ^†††^	−33.9 (−54.3; +7.2)^†^	0.7	0.7
C-peptide (pmol/L)	1342 (1001; 1643)	−152 (−327; −2) ^†††^	−250 (−334; −42.4) ^†††^	1256 (1028; 1617)	−170 (−286; 4) ^†††^	−193 (−354; −62) ^†††^	0.9	0.6
HOMA2 IR	3.5 (2.7; 4.3)	−0.5 (−1.2; −0.2) ^†††^	−0.8 (−1.2; −0.3) ^†††^	3.5 (2.7; 4.1)	−0.7 (−1.0; −0.2) ^†††^	−0.7 (−1.0; −0.2) ^†††^	0.6	1.0
HOMA2 B%	71.1 (51.0; 98.5)	20.6 (10.8; 33.6) ^†††^	9.0 (2.0; 17.2) ^††^	79.0 (55.5; 95.0)	27.3 (12.0; 41.9) ^†††^	7.3 (−3.5; 7.6) ^†^	0.7	0.3
Lipid metabolism								
Total cholesterol (mmol/L)	3.9 (±0.8)	−0.5 (±0.6) ^†††^	0.2 (±0.6)	3.7 (±0.9)	−0.6 (±0.6) ^†††^	0.3 (±0.8)	0.9	0.4
HDL cholesterol (mmol/L)	1.1 (±0.2)	−0.1 (±0.1) ^††^	0.2 (±0.1) ^†††^	1.2 (±0.3)	−0.1 (±0.2) ^†^	0.2 (±0.2) ^†††^	0.3	0.3
LDL cholesterol (mmol/L)	2.2 (±0.7)	−0.3 (±0.4) ^†††^	0.1 (±0.5)	2.0 (±0.7)	−0.4 (±0.4) ^†††^	0.1 (±0.6)	0.9	0.2
Non-HDL cholesterol (mmol/L)	2.8 (±0.8)	−0.4 (±0.6) ^†††^	0.0 (±0.6)	2.5 (±1.0)	−0.5 (±0.6) ^†††^	0.0 (±0.8)	0.8	0.2
Triglycerides (mmol/L)	1.7 (1.3; 2.1)	−0.3 (−0.5; 0.1) ^†^	−0.2 (−0.5; 0.2)	1.5 (1.1; 2.1)	−0.4 (−0.7; −0.1) ^†††^	−0.3 (−0.5; 0.1) ^†^	0.6	0.2
NEFA (mmol/l)	0.70 (0.65; 0.84)	−0.1 (−0.2; 0.0)	−0.3 (−0.4; −0.1) ^†††^	0.60 (0.50; 0.70)	−0.0 (−0.1; 0.1)	−0.2 (−0.3; −0.1) ^†††^	0.8	0.7
ApoA-I (g/L)	1.34 (±0.2)	−0.13 (±0.14) ^†††^	0.14 (±1.6) ^†††^	1.39 (±0.26)	−0.15 (±0.19) ^†††^	0.17 (±0.18) ^†††^	0.4	0.2
ApoB (g/L)	0.85 (±0.20)	−0.10 (±0.12) ^†††^	−0.00 (±0.13)	0.77 (±0.22)	−0.13 (±0.13) ^†††^	0.00 (±0.17)	0.7	0.3
ApoB/ApoA-I	0.65 (0.55; 0.74)	−0.02 (−0.10; +0.06)	−0.07 (−0.10; −0.01) ^††^	0.50 (0.43; 0.64)	−0.04 (−0.08; −0.01) ^†^	−0.05 (−0.10; −0.01) ^†††^	0.4	0.5
Kidney function								
eGFR (ml/min/1.73 m^2^)	78.0 (±17.0)	−1.3 (±6.5)	−5.2 (±7.0) ^†††^	80.2 (17.3)	−0.2 (±6.6)	−5.1 (±4.6) ^†††^	0.9	0.3
Albumin (μmol/L)	0.08 (0.03; 0.16) ^A^	−0.02 (−0.07; 0.00) ^A^	−0.00 (−0.06; 0.00) ^A^	0.07 (0.03; 0.19) ^B^	−0.01 (−0.05; 0.00) ^B^	−0.00 (−0.05; 0.01) ^B^	0.7	0.3

Values are expressed as mean (±SD) or median (quartile 1; quartile 3). Baseline values are expressed as absolute values. Week 6 and 30 values are indicated as changes from baseline values. Descriptive statistics include only participants with complete datasets. Differences between diets were assessed using linear mixed models using all available data. Residuals were tested for normal distribution and log transformation was used when deemed appropriate. *p*1 value expresses statistical significance in changes between diets from baseline to week 30 while adjusting for baseline. *p*2 value expresses significance in changes between diets from week 6 to week 30 without baseline adjustment. Significance from baseline (within group); ^†^
*p* < 0.05, ^††^
*p* < 0.01, ^†††^
*p* < 0.001. Participants supplied the urine samples used in this statistic; ^A^
*n* = 28, ^B^
*n* = 24.

**Table 4 nutrients-14-05244-t004:** Medication changes.

	CD (*n* = 31)		CRHP (*n* = 28)		*p*-Value
	Increased Dose	Decreased Dose	Increased Dose	Decreased Dose	
Anti-hyperglycemic	5	5	4	7	0.7
Lipid-lowering	3	0	0	0	0.2
Anti-hypertensive	3	1	3	2	0.9

Values are expressed as frequency counts (number of patients who increased or decreased medication). Differences between groups are assessed using Fisher’s exact test including participants who had no changes in medication. Only participants who completed the entire study are included in this analysis.

## Data Availability

The dataset generated during this study is available upon reasonable request from the corresponding author and approval by the Danish Data Protection Agency.

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
