# Peer review of "Effects of Carbohydrate Restriction on Body Weight and Glycemic Control in Individuals with Type 2 Diabetes: A Randomized Controlled Trial of Efficacy in Real-Life Settings"

_nutrients, 2022, doi:10.3390/nu14245244_

Round 1

Reviewer 1 Report

Thank you for the opportunity to review this interesting manuscript.

An issue of interest is whether dietary adherence during self-prepared food intake after 6 weeks of full diet provision is achieved by some and not others, or whether everyone fails to comply to different degrees – I appreciate that the trial set-up is not appropriate to fully investigate this, but are the authors able to comment?  

The authors emphasise (line 276) the importance of weight loss rather than dietary macronutrient composition in the long-term management of T2DM, and conclude that future studies should address questions of how to improve long term adherence to low carbohydrate diets. I am surprised the authors did not suggest a long-term trial with full diet provision – no doubt this would be a large undertaking but is there is simple summary point for why this would be impractical?  

Minor comment:

Table 2 – the sign on the point estimates for between diet difference for CHO (%E) and fibre (g/day) should be negative, shouldn’t they? – these may have been lost during table production.

Author Response

Thank you for the opportunity to review this interesting manuscript.

An issue of interest is whether dietary adherence during self-prepared food intake after 6 weeks of full diet provision is achieved by some and not others, or whether everyone fails to comply to different degrees – I appreciate that the trial set-up is not appropriate to fully investigate this, but are the authors able to comment?  

Only a small handful of participants were able to achieve (approximately) targeted macronutrient distributions in each group. We did not systematically collect data to evaluate why some participants succeeded more than others. Having spoken to most of the participants I am left with the impression that the magnitude of the task exceeded participants resources in the long term. Many participants changed strategy (during the follow-up) to simply “reduce carbohydrate” and “increase protein” and focused less on the specific energy percentage targets.

The authors emphasise (line 276) the importance of weight loss rather than dietary macronutrient composition in the long-term management of T2DM, and conclude that future studies should address questions of how to improve long term adherence to low carbohydrate diets. I am surprised the authors did not suggest a long-term trial with full diet provision – no doubt this would be a large undertaking but is there is simple summary point for why this would be impractical?  

We do believe a long-term food provision to be an effective method to facilitate high diet adherence. Conducting the present study, we experienced multiple “barriers” for patients to live with full food provision during the initial 6-week food provision period. The most prominent “barrier” being that eating/dining is a key component in a broad range of social interactions in modern society.

Minor comment:

Table 2 – the sign on the point estimates for between diet difference for CHO (%E) and fibre (g/day) should be negative, shouldn’t they? – these may have been lost during table production.

Answer:

Thank you for this observation. Plus (+) and minus (-) signs are added to correctly indicate the difference in change over time to each endpoint in Table 2.

Reviewer 2 Report

This is camera ready.  Everything about this paper is stellar. BRAVO

Author Response

This is camera ready.  Everything about this paper is stellar. BRAVO

Answer:

Thank you!

Reviewer 3 Report

Title: Effects of Carbohydrate Restriction on Body Weight and Glycaemic Control in Individuals with Type 2 Diabetes: A Randomized Controlled Trial of Efficacy in Real-Life Settings

Comments to Authors  

Although the study looks interesting, there are some issues with the following findings:

1. Introduction: 

-The authors should clarify the novelty of this article in the ‘Introduction’ and ‘Conclusion’ section.

2. Is there any evidence of in vitro or animal model studies? If this is the case, you must discuss it in the introduction and discussion sections. If not, how did you get involved in human research?

3. Did you find any minor/major adverse side effects with these diet during the studies period. If any author has to discuss in the discussion section.

4. Results  

- The presentation of Tables 2 and 3 is in poor quality. It's quite perplexing. I would advise making it more precise so that the reader may easily grasp it.

- There was a high probability of assessing glucose tolerance and insulin sensitivity, as well as beta cell and liver function. It has potential to increase the scientific value of this paper. Why did the author not bother? If it is possible, I would recommend carrying out these tests.

5. Discussion:

-Discussion section is the paramount of introduction. However, it sounds okay as per their studies, but it gives the reader the opportunity to raise a question because

 -It has very limited number of patients (sample size very smaller)

-Co-relation with previous studies is very limited.

6. How is this article more informative than the previously published ones? Justify it.

7. Conclusion:

-This section does do not sound good. Author has to rewrite this section focusing on their outcomes corelating with future prospective.

Author Response

Thank you for your comments on our manuscript. Please see the attatchment for a reply to each one of them.

Thank you for the comments on our manuscript please find separate answers for each comment below.

  1. Introduction: 

-The authors should clarify the novelty of this article in the ‘Introduction’ and ‘Conclusion’ section.

answer

We consider the manuscript a puzzle piece of a larger story on carbohydrate restriction and protein enrichment. This novelty of the study should be evaluated in the larger context of the series of trials named “CutDM”. We have reworked the introduction and conclusion to emphasize this.

The introduction section (line 56-62) have been changed from:

“More recently, we demonstrated that a fully-provided, hypocaloric CRHP diet for 6 weeks improves glycemic control and reduces liver fat content to a greater extent than the same amount of weight loss achieved by a CD diet (11). In this follow-up study, we report on the results of the subsequent 24 weeks of free-living follow-up with a less frequent dietitian guidance on complying to allocated diet patterns but visits only every other month.”

The introduction section (line 56-62) has been changed to:

“More recently, we demonstrated that the same CRHP diet ameliorated the beneficial metabolic effects of weight loss when applied during calorie restriction in a fully controlled setting for 6 weeks (11). In this follow-up study, we report on the results of the subsequent 24 weeks of free-living diet intervention to investigate if the additive metabolic effect of calorie restriction and carbohydrate reduction could be maintained when patients self-prepared diets with the aid of dietitian guidance.”

  1. Is there any evidence of in vitroor animal model studies? If this is the case, you must discuss it in the introduction and discussion sections. If not, how did you get involved in human research?

Answer:

There is a recent animal study[i] of 6 weeks where 30E% of carbohydrate has been substituted by fat, which then took up 65E% of the diet. In line with ours 6 weeks studies of full provision of diet with reduced carbohydrate, that animal study showed a halving of liver fat and plasma triglyceride. We do not believe animal studies have piloted modern human research in dietary treatment of Type 2 diabetes, and therefore choose not to include this in the introduction/discussion. Experimentation in human nutritional distribution of macro nutrients has been of increasing interest for half a decade. Our focus lies on the treatment of humans in the light of the worldwide increasing diabetes mellitus epidemic.

  1. Did you find any minor/major adverse side effects with these diet during the studies period. If any author has to discuss in the discussion section.

Answer:

Adverse events were only registered in the controlled 6 week period and have been published earlier (Thomsen et al. 2022), as indicated in the results section (section 3.1.).

Thirteen participants (5 in the CD group and 7 in CRHP group) experienced mild constipation symptoms, while one participant in the CRHP group experienced severe constipation. 2 participants in each group experienced diarrhea. Three participants (1 in the CD group and 2 in the CRHP group) experienced dizziness. One participant experienced a transient episode of excessive sweating combined with increasing levels of plasma creatinine; however, no underlying medical cause was identified.

  1. Results  

- The presentation of Tables 2 and 3 is in poor quality. It's quite perplexing. I would advise making it more precise so that the reader may easily grasp it.

Answer:

I understand and agree to the comment regarding the layout of the tables mentioned. It seems the tables were distorted in the editing process of applying the papers official template. I apologies for this and have made adjustments in the new edition which hopefully makes the information more presentable. See Table 2 and 3.

- There was a high probability of assessing glucose tolerance and insulin sensitivity, as well as beta cell and liver function. It has potential to increase the scientific value of this paper. Why did the author not bother? If it is possible, I would recommend carrying out these tests.

Answer:

I agree insulin sensitivity and beta cell function as well as liver fat fraction are of very high interest in this study design. In the initial 6-week food provision period of the study insulin sensitivity and beta cell function was assessed using Oral glucose tolerance test and hepatic fat fraction was assessed using magnetic resonance imaging. These results have been published previously (Thomsen et al. 2022) as indicated in the text. Unfortunately, these high-quality tests were not available at the 30-week follow-up. As a replacement, insulin sensitivity and beta cell function was assessed using HOMA-2 modelling as described in the methods and materials section. The results did not reveal any significant changes between dietary groups (table 3).

  1. Discussion:

-Discussion section is the paramount of introduction. However, it sounds okay as per their studies, but it gives the reader the opportunity to raise a question because

 -It has very limited number of patients (sample size very smaller)

-Co-relation with previous studies is very limited.

Answer:

1: This follow-up study should be interpreted in the relation to the studies we have conducted previously, on the nature of carbohydrate restriction and protein enrichment as treatment for patients who suffer from type 2 diabetes.

2: We agree that studies with larger population size would give the research more power. The study size was calculated using basic statistical power calculation ensuring a power level of 80% with a level of significance at 5% to reject the null hypothesis, based on results from one of our own previous studies of resembling design.

3: Studies addressing low carbohydrate and high protein diets are very heterogeneous in design making comparison very difficult. In this follow-up with self-selected and self-prepared diets we also seek to investigate the applicability of the defined diets to a real-world setting.

  1. How is this article more informative than the previously published ones? Justify it.

Answer:

We have conducted a series of (so far) 3 studies in the CutDM series, which, each are designed to provide information on different aspects of carbohydrate restriction in the treatment of type 2 diabetes mellitus.

The present study sought to explore if participants would be able to follow a carbohydrate reduced diet pattern in a real world setting as we have seen earlier (Alzahrani et al. 2021), but with the complexity of also having to maintain a weight loss during the study period.

  1. Conclusion:

-This section does do not sound good. Author has to rewrite this section focusing on their outcomes corelating with future prospective.

Answer:

We have revised the conclusions section accordingly.

The conclusion section is changed from:

“Although carbohydrate restriction beneficially affects glucoregulation in individuals with T2DM in a manner that goes beyond weight loss under tightly controlled conditions (full diet provision), any additional effects were attenuated after 24-week of diet self-preparation in a real-life setting. Weight loss and its maintenance was shown to be instrumental in improving glycemic control in T2DM. Future studies should address the key question of how to improve long-term adherence to carbohydrate-restricted dietary regimens.”

The conclusion section now reads:

“The additive metabolic effects of carbohydrate reduction from weight loss were lost in the present extension study, when participants transitioned from full food provision to a real-life setting. This contrasted our previous findings from carbohydrate restriction without accompanying weight loss, maybe because participants were unable to simultaneously adhere to the allocated diet pattern and keep their weight reduction. Maintenance weight losses was shown to be instrumental in improving glycemic control in T2DM. Future studies must address how to improve long-term adherence to carbohydrate-restricted dietary regimens to determine the possible positive effect of such diets in improving several pathophysiological traits of type 2 diabetes.”

[i] Lundsgaard et al., Mechanisms Preserving Insulin Action during High Dietary Fat Intake. Cell Metab 29, 50-63 e54 (2019).

Round 2

Reviewer 3 Report

1. The majority of the comments have been addressed by the authors. However, the following minor changes are required.

2. I highly suggest the author to add adverse side effects with these diets during the trial period and explain the reasons behind the limitations of the study in the discussion section.

Author Response

The majority of the comments have been addressed by the authors. However, the following minor changes are required.

I highly suggest the author to add adverse side effects with these diets during the trial period and explain the reasons behind the limitations of the study in the discussion section.

Answer

Section 3.1 following change has been made to inform about  adverse  effects of the study interventions:

The following was deleted:

“Adverse events in the first 6 weeks have been reported elsewhere (11) and did not lead to participant withdrawal.”

And replaced with the following sentences:

“Adverse events were registered during the 6-week food provision period only. Thirteen participants (5 in the CD group and 7 in CRHP group) experienced mild constipation symptoms, while one participant in the CRHP group experienced moderate constipation. Two participants in each group experienced diarrhea. Three participants (1 in the CD group and 2 in the CRHP group) experienced dizziness. One participant experienced a transient episode of sweating and increased levels of plasma creatinine; no underlying medical cause was identified (11).”

The following paragraph in the discussion section has been changed to include reasons behind study limitations:

The following was deleted:

“This follow-up study has several limitations, including the open-label design, the short duration relative to the timeframe of weight loss maintenance, the lack of validated methods to account for medication changes during follow-up, and providing group-based rather than individual dietetic advice. Moreover, the frequency of dietitian guidance was low compared to other studies with similar objectives.”

The following was supplied:

“This follow-up study has several limitations; open-label design which is a natural consequence of dietary studies. The short study duration,  group-based rather than individual dietetic advice, and the relatively low frequency of dietitian guidance were chosen to comply with the limited study resources. The lack of accurate methods to account for changes in medication during follow-up may have clouded differences in the metabolic effect between dietary groups.”